# The Effect of Upgrades to Childcare Outdoor Spaces on Preschoolers’ Physical Activity: Findings from a Natural Experiment

**DOI:** 10.3390/ijerph17020468

**Published:** 2020-01-10

**Authors:** Michelle Ng, Michael Rosenberg, Ashleigh Thornton, Leanne Lester, Stewart G. Trost, Pulan Bai, Hayley Christian

**Affiliations:** 1Telethon Kids Institute, University of Western Australia, Perth 6009, Australia; pulan.bai@telethonkids.org.au (P.B.); hayley.christian@uwa.edu.au (H.C.); 2School of Human Sciences M408, University of Western Australia, Crawley, Perth 6009, Australia; michael.rosenberg@uwa.edu.au (M.R.); ashleigh.thornton@uwa.edu.au (A.T.); leanne.lester@uwa.edu.au (L.L.); 3Institute of Health and Biomedical Innovation, Centre for Children’s Health Research, Queensland University of Technology, Brisbane 4101, Australia; s.trost@qut.edu.au; 4School of Population and Global Health M431, University of Western Australia, Perth 6009, Australia

**Keywords:** early childhood, preschool, childcare, physical activity, outdoor environment, built environment, natural experiment

## Abstract

Physical inactivity is a significant risk factor for childhood obesity. Preventing obesity in the early years reduces the risk of developing chronic health conditions later. Early childhood education and care (ECEC) services are important settings to establish good preschooler physical activity behaviors. This natural experiment investigated the influence of ECEC outdoor physical environment upgrade on preschoolers’ physical activity (aged 2–5 years). Centers implemented upgrades without researcher input. Physical activity was measured by 7-day accelerometry for intervention (n = 159; 6 centers) and control (n = 138; 5 centers) groups. ECEC outdoor space was assessed using a modified Environment and Policy Assessment and Observation (EPAO) Instrument. Key outcomes were measured at baseline and 6–12 months follow-up. Fixed sandboxes, balls, portable slides, portable floor play equipment (e.g., tumbling mats), and natural grassed areas were positively associated with activity levels; fixed tunnels and twirling equipment were negatively associated with activity levels (all *p* < 0.05). Post-upgrade portable play equipment (balls, twirling equipment, slides, floor play equipment) increased intervention preschoolers’ moderate-vigorous physical activity (MVPA) levels compared to control (*p* < 0.05). Intervention preschoolers were more active than control at follow-up (58.09 vs. 42.13 min/day increase in total physical activity; 30.46 vs. 19.16 min/day increase in MVPA (all *p* < 0.001)). Since few preschoolers meet daily activity recommendations while at ECEC, the findings may help ECEC providers to optimize outdoor physical environments and encourage more active play among preschoolers.

## 1. Introduction

The number of children attending some form of early childhood education and care (ECEC) in most developed countries has increased dramatically in recent decades [1]. According to the OECD (Organization for Economic Co-operation and Development), the number of Western European children enrolled in ECEC increased from 20% to 90% over a 15–20 year period between 1994 and 2014 [2]. In Australia, 54% of 2–3-year-olds and 85% of 4–5-year-olds attend ECEC [3]; in the United States, 40% of 3-year-olds are enrolled in ECEC [2]; in Canada, 54% of children aged 4 years and under are enrolled [4]; and in France, 100% of 3-year-olds are enrolled [2].

In 2015, excess body weight resulted in about 4 million deaths and 120 million disability-adjusted life-years worldwide [5]. Promoting physical activity behaviors in early childhood is crucial to preventing obesity and is an international priority [6]. Childhood overweight and obesity are associated with an increased risk of developing several preventable health conditions, such as cardiovascular disease, diabetes, depression, arthritis, and premature mortality [7]. In addition, overweight and obese children are more likely to remain obese as they enter adulthood [8]. Regular physical activity mitigates such risks and is also associated with various positive physical health outcomes (such as improved bone health and cardiovascular fitness), socio-emotional development (such as enhanced social skills and emotional intelligence), mental health (such as reduced depression and anxiety problems), and better sleep in children [9,10,11]. Early childhood is a critical time for establishing healthy patterns of wellbeing and physical activity behaviors [1,12,13]; therefore, there is a significant opportunity to influence behaviors of young children while attending ECEC [14].

Recently, the World Health Organization (WHO, Geneva, Switzerland) and countries such as Australia, Canada, and New Zealand released 24-h movement guidelines for the early years [15,16,17,18]. The Guidelines recommend children aged 2–5 years should spend at least 180 min in a variety of physical activity each day, of which 60 min is energetic play. Estimates of the proportion of young children meeting physical activity guidelines vary considerably. In Australia, 34% of preschoolers achieve recommended physical activity levels each day [19]. In Canada, 62% of three to four-year-old preschoolers meet physical activity guidelines [20]. These differences in the proportion of preschoolers meeting physical activity guidelines could be due to differing methodologies (subjective vs. objective measures), in particular, different cut-points for light-intensity physical activity [21]. Despite this, most studies show that preschoolers accumulate most of their physical activity in low-intensity activity [22,23,24] and a significant proportion do not meet the guidelines.

Outside of the home, ECEC is an important setting to promote young children’s physical activity. Correlates of preschoolers’ physical activity while at ECEC include child age (older children are often more active), sex (boys are often more active), better fundamental motor coordination, educator behaviors (i.e., prompts and feedback), and the provision of active opportunities for physical activity and certain features of the outdoor environment (e.g., size, use of and presence of portable play equipment like bikes and balls), and these correlates have all shown positive associations [1]. Modifying the outdoor physical environment has significant potential to influence preschoolers’ physical activity while attending ECEC [1]. However, there are few intervention studies involving changes to the ECEC outdoor physical environment as such studies are difficult and expensive to implement [25]. Findings from these intervention studies have been mixed. A recent Canadian study found that adding novel portable equipment, in addition to staff training and modifying outdoor playtime, increased objectively measured physical activity levels in children attending the ECECs that had been modified [26]. A similar finding was reported by a US study, which found that adding portable play equipment to the ECEC playground increased children’s physical activity [27]. Another US study found adding pathways and improving the overall quality of the physical environment increased physical activity levels in children [28]. Conversely, a Belgian study found adding portable play equipment and playground markings were not positively associated with physical activity [29]. The mixed findings from previous studies could be attributed to methodological differences including differing study designs, small sample sizes, subjective measures of children’s physical activity, and short follow-up periods. More rigorous intervention studies are required to address the evidence gap and better understand how changes to the ECEC outdoor physical environment impacts upon children’s activity levels.

Due to the high expense and difficulty in implementation, no such studies have been conducted in Australia. Natural experiments using quasi-experimental research designs have been identified as a research priority for demonstrating the causal relationships between built environments and physical activity [30]. The Western Australian Play Spaces and Environments for Children’s Physical Activity (PLAYCE) study provided a unique opportunity to conduct a natural experiment in ECEC. This study had two aims: (1) to measure the change in the ECEC outdoor physical environment and (2) examine the impact of changes to the ECEC outdoor physical environment on children’s physical activity levels while attending care.

## 2. Materials and Methods

### 2.1. Study Design and Participants

This was a sub-study of the Play Spaces and Environments for Children’s Physical Activity (PLAYCE) study. The PLAYCE cross-sectional study investigated the relative and cumulative influence of the ECEC, home, and neighborhood environment on preschoolers’ physical activity. Full details of the PLAYCE study methods have been published [31]. Briefly, between 2015 and 2017, 1596 preschoolers aged 2–5 years and their parents were recruited from 104 ECEC services across metropolitan Perth, Western Australia [19]. The sampling and recruitment of centers were stratified by socioeconomic areas (low, medium and high) and size of center [31]. Tiered consent was utilized, requiring consent from center directors first before parental consent (response rate 24%). Preschoolers were excluded if they were attending full-time school and had any intellectual, emotional, physical, or behavioral disabilities, which hindered participation in physical activity. An ethics amendment to the PLAYCE study for this natural experiment study was granted by The University of Western Australia Human Research Ethics Committee (#RA/4/1/7417).

This current study was an opportunistic natural experiment conducted between 2016 and 2017. The study used a pre-post-test design with intervention and matched control centers to evaluate the effect of changes to the ECEC outdoor physical environment on children’s physical activity behavior while at ECEC. Centers were invited to join the intervention group if they informed the study team that they were undertaking a major upgrade of their outdoor physical environment within six months of their first data collection, which was conducted as part of the main study. This was considered the baseline assessment for the current study. After intervention centers were recruited, control centers were then selected based on two criteria: (1) had their baseline assessment completed between July and December 2016, and (2) matched one-to-one to intervention centers based upon SES tertile (low, middle, or high). Matching of centers on center SES was done to ensure that intervention and control centers had access to similar funding and thereby should have similar outdoor physical environment features at baseline. Once centers provided consent to a follow-up assessment of their center, all parents within the center were invited to participate, i.e., parents who had previously participated in the PLAYCE study and parents who had not previously participated in the PLAYCE study. The current study was an opportunistic pilot natural experiment nested within the larger PLAYCE cross-sectional study; it was not the original design of the PLAYCE study to evaluate the impact of ECEC interventions on children’s physical activity. Therefore, not all children who completed the baseline assessments as part of the PLAYCE study were available for follow-up assessment as part of the current study, the main reasons for this included children leaving the center to commence kindergarten or full-time school, or parent unwillingness to re-participate. Therefore, ‘new children’ were recruited to the study at follow-up to ensure a sufficient sample size.

Six intervention centers (4 middle SES, 2 low SES; 4 large size, 1 middle size, and 1 small size) with 159 intervention children and 5 control centers (3 middle SES, 2 low SES; 3 large size, 1 middle size, and 1 small size) with 138 children participated (Figure 1). As control centers were recruited after the baseline assessment was completed, only five centers met the selection criteria to be included as control centers: two intervention centers from the middle SES tertile were matched to one control center of the same SES tertile.

### 2.2. Instruments

#### 2.2.1. Physical Activity

Physical activity was objectively measured using ActiGraph GTX3+ (ActiGraph, Pensacola, US) accelerometers, which have demonstrated validity for measuring physical activity in young children [32]. A 15 s sampling interval (epoch) to accommodate the typical nature of children’s physical activity was used. Physical activity intensity was classified based on the following cut points developed by Pate and colleagues: sedentary (<200 counts/15 s), light intensity (200–419 counts/15 s), and moderate to vigorous intensity physical activity (MVPA) (420 counts/15 s) [33]; the cut points have been validated for use in preschoolers by Trost and colleagues [34]. Non-wear time was defined as strings of consecutive zero counts lasting 20 min or longer. ECEC monitoring days were considered valid based on at least 1 day at ECEC with 75% wear time [31]. Wear time was calculated by taking non-wear time from monitoring time. Accelerometers were worn on the right hip for 7 days during waking hours. Parents reported the days and times their child attended ECEC in the accelerometer diary. Only accelerometer data for the time children were at ECEC were analyzed for which there was a minimum of at least one day at day ECEC with 75% wear time [31].

A diary was used to record if the accelerometer was removed, the amount of time the accelerometer was worn and the days, and times children attended ECEC. Accelerometer data were analyzed using a custom-built excel and SAS macro developed for analyzing the intensity and amount of physical activity children accumulate while attending ECEC and overall. Data for the average minutes of total physical activity (TPA) (calculated as sum of light physical activity and MVPA) and MVPA per ECEC day were used in analyses.

#### 2.2.2. ECEC Outdoor Physical Environment

The main PLAYCE study measured the ECEC outdoor physical environment using a slightly modified version of the Environment and Policy Assessment and Observation (EPAO) Instrument [35] ‘Physical Environment’ subscale to ensure relevance for the Australian context [31]. The EPAO instrument was developed to assess the quality of ECEC centers physical activity and eating behaviors in the US [35,36] and has been validated [37,38]. The revised tool has been previously described in the PLAYCE study methods paper [31]. The audit tool assessed the presence of physical activity opportunities in the physical environment [39]. The revised audit items have been shown to have excellent intra-class correlation coefficients (ICCs) for intra-rater reliability, and good to excellent ICCs for inter-rater reliability. For intra-rater reliability of the outdoor physical environmental audit, the ‘Fixed play equipment’ subscale ICC was 0.94, the ‘Portable play equipment’ subscale was 0.94, the ‘Natural physical features’ subscale was 0.84, and the ‘Outdoor play spaces’ subscale was 0.80. For inter-rater reliability of the outdoor physical environmental audit, the ‘Fixed play equipment’ subscale ICC was 0.7, the ‘Portable play equipment’ subscale was 0.74, the ‘Natural physical features’ subscale was 0.79, and the ‘Outdoor play spaces’ subscale was 0.79.

The current study included data from five outdoor physical environment subscales in the audit tool: ‘Fixed play equipment’ and ‘Portable play equipment’ from the EPAO [35], ‘Total size of playing area’ [40], ‘Outdoor play spaces’ [41,42,43], and ‘Natural elements’ [41,42,43]. The physical environment subscales were created using the scoring tool and guidelines provided by the original authors. Each subscale was created at the child level. While some intervention centers had more than one outdoor area that a child could access, upgrades were implemented in all outdoor areas, therefore, all children were exposed to the upgrade intervention.

The ‘Fixed play equipment’ subscale score was based on the availability of eight types of equipment: structured tracks (e.g., playground markings), climbing structures (e.g., jungle gyms), see-saws, slides, tunnels, balancing surfaces (e.g., balance beams), sandboxes, and swinging equipment (e.g., swings, ropes). Merry-go-rounds measured in the EPAO were not included in analyses as none of the ECEC outdoor environment had this feature. Items were coded 1 if present and 0 if not; total score for ‘Fixed play equipment’ was calculated as the sum, divided by eight (number of items) and multiplied by 10 (to obtain a score from 0 to 10, higher score indicated more activity opportunities). The ‘Portable play equipment’ subscale score was calculated in a similar manner, and included nine items: balls, climbing structures (e.g., ladders), floor play equipment (e.g., tumbling mats), jumping equipment (e.g., jump ropes, hula hoops), push/pull toys (e.g., wagons), riding toys (e.g., tricycles, cars), slides, sand/water toys (e.g., buckets, scoops), and twirling equipment (e.g., ribbons, batons). ‘Total playing area’ was rated on a scale from 0 (no playing area) to 10 (very large area) by comparing all ECEC outdoor playing areas and dividing them into 10 tertiles. The ‘Natural elements’ subscale score was calculated in a similar manner and included eight items: mature trees, other plants, vegetable/herb/fruit garden, rocks/stones/pebbles, natural grassed area, artificial grassed area, potted plants, and flower beds. The ‘Outdoor play spaces’ subscale score was also calculated in a similar fashion, and included five items: open areas, water play areas, sloping ground, a variety of ground surfaces (e.g., mulch, artificial covering), and playground constructed at different levels.

Research personnel conducting the environmental audits were trained in using the EPAO. A senior research assistant was present on site to answer any queries and ensure consistency. Intra-rater and inter-rater reliability intraclass correlations (ICCs) were good to excellent (ICC = 0.70–0.94).

### 2.3. Confounders

Child age, sex, and parental education variables were collected via established items in the PLAYCE parent survey [31].

### 2.4. Statistical Analysis

All analyses were conducted using SPSS V.24, *p* values < 0.05 were considered statistically significant. Analyses were conducted at the child level to adjust for child-related confounders (age, sex, parent education, and accelerometer wear time). All data were analyzed using the intention-to-treat principle.

Descriptive statistics were used to examine various background characteristics. The significance of differences in preschooler activity levels between groups and within groups was examined using *t*-tests. The distribution of the outdoor physical environment measures was explored and the significance of differences between features at baseline and follow-up were examined using t-tests. Effect sizes were interpreted using the classification defined by Cohen [44].

Multiple multivariate linear regression analyses with three levels (i.e., child level; room level; center level) were conducted to examine the associations between the ECEC outdoor physical environment features and child characteristics as independent variables, and physical activity levels (TPA and MVPA) as outcome variables, and random intercepts at the child, room, and center level. Analyses were conducted with time (baseline and follow-up), group (intervention or control), and the interaction between group and time as covariates. Models were run separately for each ECEC outdoor physical environment subscale: ‘Fixed play equipment’ and ‘Portable play equipment’, ‘Total size of playing area, ‘Outdoor play spaces’ and ‘Natural elements’. Insignificant independent variables were stepwise deleted from the model in order of their significance, starting with the least significant variable. This procedure was repeated until all remaining independent variables were significant.

## 3. Results

### 3.1. Participant Baseline Characteristics

At baseline, the average age of participants was 2 years 10 months (±SD = 0.82) and 51.1% were girls (results not shown). There were no significant differences between intervention and control group sociodemographic characteristics at baseline except that the intervention group had fewer parents with a bachelor’s degree or higher qualification compared with the control group (44% vs. 59%, *p* = 0.04).

### 3.2. Child Physical Activity Level

Table 1 describes child physical activity levels for both intervention and control groups at baseline and follow-up. At baseline, on average, children accumulated 63.67 min (±SD = 64.68) of TPA and 31.17 min (±SD = 34.82) of MVPA per average day at ECEC. At baseline, intervention children accumulated significantly less TPA (58.76 min (±SD = 63.16) vs. 68.67 min (±SD = 66.23)) and less MVPA than control children (28.43 (±SD = 34.03)) vs. (34.32 (±SD = 35.58)). A very low proportion of children accumulated the recommended 180 min of physical activity per average ECEC day with intervention children accumulating significantly less (intervention 4.4% vs. control 5.8%).

All physical activity differences within groups were significant (*p* < 0.05) with a large effect size in favor of follow-up (d between 0.86 and 1.70) (Table 1). Children in both groups accumulated more activity at follow-up, but it was higher in the intervention group (TPA and MVPA). No significant effects were found for between-group differences for change in activity levels between follow-up and baseline.

### 3.3. ECEC Outdoor Physical Environment Features

Table 2 provides an overview of changes in ECEC outdoor physical environment features between intervention and control groups, and within groups between baseline and follow-up. At baseline, the intervention group had higher scores for all subscales compared with the control group. Environmental scores for all subscales, except the control group’s ‘Total Outdoor Playing Area’ subscale score, decreased at follow-up for both intervention and control groups. All within-group differences, except ‘Total Outdoor Playing Area’, between baseline and follow-up measurements for intervention and control groups were significant (*p* < 0.05). All effect sizes were in favor of baseline: the intervention group had a larger effect size than the control group (d = −0.95 to −0.79 vs. −0.77 to −0.43). Between-group effect sizes were small (d between −0.12 to −0.32) and in favor of the control group. Significant mean changes were found for ‘Portable Play Equipment’ and ‘Outdoor Play Spaces’ subscale scores (*p* < 0.05): for both subscales, the control group’s score decreased less than the intervention group’s.

### 3.4. ECEC Outdoor Environment Features and Physical Activity

Table 3 shows the associations of the ECEC outdoor physical environment and child background factors with outdoor physical activity levels. Fixed sandbox (Model 1), balls (Model 2), and presence of natural grassed area (Model 3) were positively associated with TPA and MVPA. Portable slides (Model 2) and portable floor play equipment (e.g., tumbling mats) (Model 2) were positively associated with MVPA only. Fixed tunnels (Model 1) and twirling equipment (Model 2) were negatively associated with TPA and MVPA. None of the ‘Outdoor Play Spaces’ features were significantly associated with children’s physical activity levels. Post-upgrade portable equipment comprising of slides, balls, twirling equipment, and floor play equipment was positively associated with more MVPA in intervention children than control children.

There are also significant age and gender effects consistently across all models. Older children accumulated more physical activity outdoors than younger children, and boys were significantly more active than girls.

## 4. Discussion

This natural experiment examined the influence of upgrades to the ECEC outdoor space on children’s activity levels. Results showed that post-upgrade portable play equipment (balls, slides, twirling equipment, portable floor play equipment like tumbling mats) had a beneficial effect on MVPA levels of the intervention group compared to the control group. This is in line with previous studies that showed that the presence of portable equipment encourages active play in children [26,37,45,46]. Portable play equipment, also known as ‘loose parts’, refers to open-ended play equipment that children can use in a variety of ways [47]. Children prefer playing equipment with moveable features as they are action-oriented (compared to static equipment) and can be used for various different functions [48]. Therefore, portable equipment encourages variation in activities over time, combatting the possible novelty effect of other new equipment such as fixed play equipment. In addition, floor play equipment such as tumbling mats can be used in a variety of ways to encourage active play like rolling or somersaulting or even provide soft landing surfaces for children to jump onto. The study also found that boys and older children tended to be more active [1,49,50].

In line with previous studies, we found several outdoor environmental features to be positive predictors of children’s activity levels. Children were more active (TPA and MVPA) when balls and a natural grassed area were present. Balls provide opportunities for children to engage in play while developing essential fundamental movement skills such as throwing and kicking [51,52,53]. Previous research has shown that the presence of natural elements encourages more activity and less sedentary behavior in children [50,54], and could provide more activity opportunities than non-natural environments [55]. However, there are studies that showed negative [42] or no association [56] between the presence of vegetation and children’s activity. It has been suggested that difference in types of natural elements being assessed could explain the conflicting results from studies [50]: functional natural elements such as grass and hills may encourage more activity [50]; conversely aesthetic natural elements such as plants and trees may obstruct open areas available for play [40,54,57]. Grassed areas can be used in a variety of ways such as running, playing with balls or games such as ‘follow the leader’. Future research into the functionality of various natural features in ECEC would be helpful to optimize future outdoor space design.

The presence of fixed tunnels and twirling equipment were negative environmental predictors of children’s activity levels. Tunnels allow for crawling/sliding movement that is not as active as running but is still important for developing fundamental movement skills. Our results suggest that children were engaged in mostly light activity when interacting with tunnels, where they could be playing hide-and-seek or imaginative play. Similarly, playing with twirling equipment could be more creative rather than physical activity promoting.

We found that portable slides and fixed sandboxes were positive predictors of children’s activity levels. This was in contrast to Gubbels et al.’s cross-sectional study that found that sandboxes and portable slides were associated with less activity in children attending childcare [40]. The authors suggested that sandboxes could have a confounding effect on children’s activity levels as their results showed a positive bivariate association between the sandbox and children’s activity levels [40]. ECEC centers with sandboxes tend to have larger budgets and larger spaces for play facilities, therefore, they are likely to also have many other outdoor features that are associated with more activity in children. Secondary chi-square tests showed that ECEC centers that had a sandbox also had many other environmental features including those which were found to be positively associated with activity levels (*p* < 0.001; results not shown). In practice, fixed sandboxes in Australian ECEC centers may also include large play equipment or see-saws/swings, so children could be moving actively between features within the sand play area.

In addition, portable slides can be placed in new and interesting locations within the play area to encourage children to interact with the equipment more, for example, within a large fixed play structure or between balancing beams. Portable slides tend to be shorter than fixed slides, so children may do more ‘laps’ running back up the stairs to slide down again. These ‘heavier type’ portable equipment have to be physically manipulated by educators, possibly also encouraging more involvement from educators when children are playing outside. Previous studies have shown positive associations between educator involvement and children’s activity levels [26,58]. Encouraging preschoolers to be more active requires an interaction between the availability of various outdoor space features and educator support. An outdoor environmental space feature on its own may not be positively associated with activity, however, when placed in proximity with other features it may offer more opportunities for play.

Intervention children accumulated significantly more activity (TPA and MVPA) at follow-up compared to baseline. This may be due to the availability of new portable play equipment such as balls, slides, floor play equipment, and twirling equipment at follow-up, which showed to have a beneficial influence on the intervention group’s activity levels. However, only a very small proportion of intervention children achieved the recommended three hours of physical activity while in ECEC at follow-up (4.44% of the intervention group; 6.25% of the control group). Considering that a large proportion of children attend ECEC services, it is concerning that only a small proportion of young children are sufficiently physically active. More focus should be given to support children in accumulating sufficient amounts of physical activity while attending ECEC. The results of this study suggest a number of physical environmental features of ECEC that could be targeted to promote more activity in children attending care. Even though intervention children’s TPA levels increased significantly at follow-up, the average TPA accumulated was still below the recommended three hours of physical activity while in ECEC [15,16].

Unexpectedly, control children’s activity levels also increased significantly at follow-up (TPA and MVPA). This could be in part due to different children participating at baseline and follow-up: 24 of control children participated at both time points, 35 of them participated only at follow-up. This may have meant that sociodemographic and physical activity differences in children participating at baseline compared with follow-up could have influenced changes in group physical activity levels regardless of any intervention effect from the environmental upgrade. Moreover, the smaller number of children participating at follow-up compared to baseline assessment (approximately 50% less children participated at follow-up for both groups) could have influenced the findings resulting in a less accurate representation of change in children’s physical activity post-intervention.

At baseline, the intervention group’s outdoor environment provided more activity opportunities than the control group’s. However, the intervention group’s subscale scores were still relatively low: none of the subscales scored more than 5 out of the total 10 points available. The low levels of activity opportunities in the intervention group’s outdoor environments suggest there was a significant opportunity for improvement via an environmental upgrade.

Environmental subscale scores of the intervention group, except ‘Total Outdoor Playing Area’, significantly changed at follow-up, however, changes were in the unexpected direction. We hypothesized intervention centers would have greater availability of activity-promoting features at follow-up, however, all subscale scores decreased at follow-up. A potential reason could be related to this being a natural experiment, i.e., centers implemented changes to the outdoor space independent of the research team. Intervention centers may have made changes to features of the outdoor space that were not associated with children’s physical activity. For example, comparison of before and after photographs taken by study personnel showed that one center replaced a climbing structure with a swing set that has been shown to be associated with lower physical activity levels in children [40]. The EPAO tool assessed the availability of activity opportunities and thus measured presence of different types of items and not number of individual items. Intervention centers could have purchased many new balls and bicycles, however, this change would not have been captured in the tool. This could explain the significantly large increase in TPA and MVPA in intervention children, even though environmental subscale scores decreased at follow-up. That is, while there may have been less different types (or variety) of activity opportunities in the ECEC outdoor space at follow-up, a sheer increase in the number of existing types of activity promoting features such as more balls and bicycles may have encouraged children to be more active. Furthermore, many of the changes to the outdoor space appeared to be part of center routine maintenance rather than a significant change to the area: ‘before and after’ photographs show instances where older equipment was replaced with newer equipment, but no major changes were made to the types of features available for children to play with. Such changes may have been viewed by centers as an ‘upgrade’ as structural changes (such as turf and structures were replaced) were made to the outdoor space, albeit there was no real increase in the number of different types of activity promoting features.

Surprisingly, environmental subscale scores for the control group, except ‘Outdoor play space’, also changed significantly at follow-up: scores at follow-up were lower than at baseline. We hypothesized that there would be no change in the outdoor physical environment of control centers between baseline and follow-up. A potential reason could be control centers may have made changes to their outdoor space between baseline and follow-up (average of 12 months) but not reported it to study personnel. In practice, ECEC centers tend to undergo an annual maintenance review that includes their outdoor space. Centers may have removed older equipment, which could, in part, explain reduced scores at follow-up.

Unexpectedly, control centers had a smaller reduction in subscale scores compared to intervention centers: the control group had a favorable but small change in subscale scores for ‘Portable play equipment’ and ‘Outdoor play space’ at follow-up compared to the intervention group. As discussed earlier, intervention centers may have removed more different types of older portable play equipment as part of their upgrade but not yet replaced them with new equipment at the time of the follow-up assessment. An apparent lack of difference in ‘Total outdoor playing area’ is consistent with the idea that given outdoor play spaces are fixed areas, it is unlikely that they would change in size significantly after an upgrade. If anything, they may possibly decrease if intervention centers added more fixed play equipment to their outdoor play space. Furthermore, although the outdoor space was measured accurately using a laser distance measure, the calculation of the actual area was difficult due to the unusual shapes of many outdoor spaces. This may have led to small measurement errors. Future studies may consider using methods such as geographic information-based systems to measure the actual size of the outdoor space. Finally, the lack of observed effect of other outdoor environment features (e.g., sloping ground, water play areas) may in part be explained by them only being present in very few ECEC centers.

Even though this study focused on how the ECEC outdoor physical environment influences children’s physical activity, future studies should investigate the impact of changes to the ECEC outdoor physical environment on different types of play as well as other health and developmental outcomes [59,60]. It may be possible to expand the existing EPAO tool to encompass a child’s development outcomes, by also examining the influences of outdoor environmental features on children’s socio-emotional development. In line with the holistic development of young children [61], ECEC services should attempt to find a balance between providing outdoor physical environment features that develop gross motor skills as well as those which encourage creative and imaginative play [59]. Furthermore, in accordance with the social–ecological framework [62], the physical activity behavior of children attending ECEC is influenced by not only the physical environment but also pedagogical intentions and effects. Future intervention studies are required to examine the interaction of physical, policy, and practice level interventions on the physical activity behavior of children while attending ECEC.

Finally, cultural differences between Australia (where the study was conducted) and the US (in which the EPAO instrument was developed) [35] mean that some features were present in the Australian ECEC environment but not captured by the EPAO instrument, and vice versa. For example, merry-go-rounds are not common in ECEC outdoor spaces in Australia but are included in the EPAO instrument. Furthermore, natural features such as nature strips/corners or chicken coops (which were installed in intervention centers as part of the environmental upgrade) were not captured in the environmental audit. Future studies could examine the influence of various emerging natural elements on children’s activity levels and developmental outcomes.

### Strengths and Limitations

A strength of this study was the natural experiment design that included intervention and control groups. Intervention studies examining changes to the built environment are difficult and expensive to implement; therefore, natural experiment is a preferred study design. Furthermore, objective measures of physical activity (accelerometry) and direct observation of physical environment features rather than less reliable methods such as self-report were used. Activity opportunities in ECEC settings were assessed by analyzing a wide range of individual features based on the existing validated observation instrument EPAO rather than summarizing facilities into a single measure.

A limitation was that study personnel could not control the type of features added during the intervention and these varied widely across centers. As a result, many of the changes made to the outdoor space were not activity promoting. Findings may also have been impacted as different children participated at baseline and follow-up. It was not the original intention of the PLAYCE study to evaluate the impact of center upgrade interventions on children’s physical activity, thus follow-up children for both intervention and control were recruited after baseline assessments were completed. Baseline physical activity levels of both groups may have impacted the result; however, intervention children accumulated more TPA and MVPA at follow-up than control children despite it being a nonsignificant increase. The study sample did not include ECEC centers from high SES areas, as intervention centers were from low and middle SES areas only. Previous studies have shown that physical activity in young children is not related to SES status [63]; therefore, the lack of representation of children from high SES areas is unlikely to have had a significant impact on the results.

In addition, the study focused only on the influence of the outdoor physical environment on children’s activity levels. However, in line with the social–ecological framework, physical activity behaviors of young children are also influenced by educators’ physical activity-related practices and ECEC physical activity-related policies. To ensure changes implemented by ECEC services are physical activity promoting, future studies could recruit one or two large ECEC providers (i.e., one provider operating many ECEC centers), instead of many independent ECEC providers. This may help research teams and ECEC providers to work together to design outdoor spaces with features that have been shown through research and practice to positively influence children’s physical activity and health. The amount of outdoor playtime provided by ECEC centers could have moderated the results. None of the ECEC centers studied had a policy on outdoor time, however, it is possible that baseline levels of educator-led outdoor play-time may have impacted results. Future research should examine the impact of environmental changes on educators’ physical activity related practices (e.g., amount of outdoor play-time, role modeling, and self-efficacy).

One of the aims of this pilot study was to test study recruitment and data collection methods.

The follow-up assessment occurred on average later for the control compared with the intervention group as control centers could only be recruited once the intervention centers were confirmed (i.e., when the upgrade was to be completed). The mean time between baseline and follow-up for intervention centers was 10.7 (SD 6.0) months and for control centers was 12.4 (SD 1.1) months. Reasons for the variation in the length of time between baseline data collection and upgrade completion included budgetary constraints, delays in obtaining quotes, plan and budget approvals and installation of the upgrade, and changes to the timeline of the upgrade installation. Lastly, the seasonal variation could have impacted baseline and follow-up measurements, however, almost all centers had their baseline and follow-up assessments in the same season; and data collection was in a similar season for matched intervention and control centers. Furthermore, Perth’s (Western Australia) Mediterranean climate means there are less extreme weather conditions impacting children’s physical activity levels at ECEC. Future studies should take into account the seasonal variation in their analyses.

## 5. Conclusions

This study is one of the first to date to examine the influence of changes to the ECEC outdoor physical environment on preschoolers’ activity levels, using an intervention design with matched controls to assess the causal relationships between the ECEC outdoor physical environment and physical activity while attending ECEC. Outdoor play spaces provide an important opportunity for children’s physical activity, play, learning, and development. The addition of new portable equipment comprising balls, slides, twirling equipment, and floor play equipment resulted in intervention children being more active at follow-up. Features such as fixed sandboxes and real grass were also found to be beneficial for activity levels. Conversely, children were observed to be less active when fixed tunnels and twirling equipment were available for play.

Even though many preschool-aged children attend and spend a large part of their waking day at ECEC, very few of them achieve the recommended activity level while at care. Findings from the current study may provide some direction for ECEC providers for optimizing their outdoor physical environment to encourage more active play among preschoolers. Holistic development of the child, including physical and socio-emotional aspects, requires a balance between ECEC physical environment features that promote gross motor development of the child, and other features that nurture creative and imaginative play. Future intervention studies are required to examine the interaction of physical, policy, and practice level interventions on the physical activity behavior of children while attending ECEC.

## Figures and Tables

**Figure 1 ijerph-17-00468-f001:**
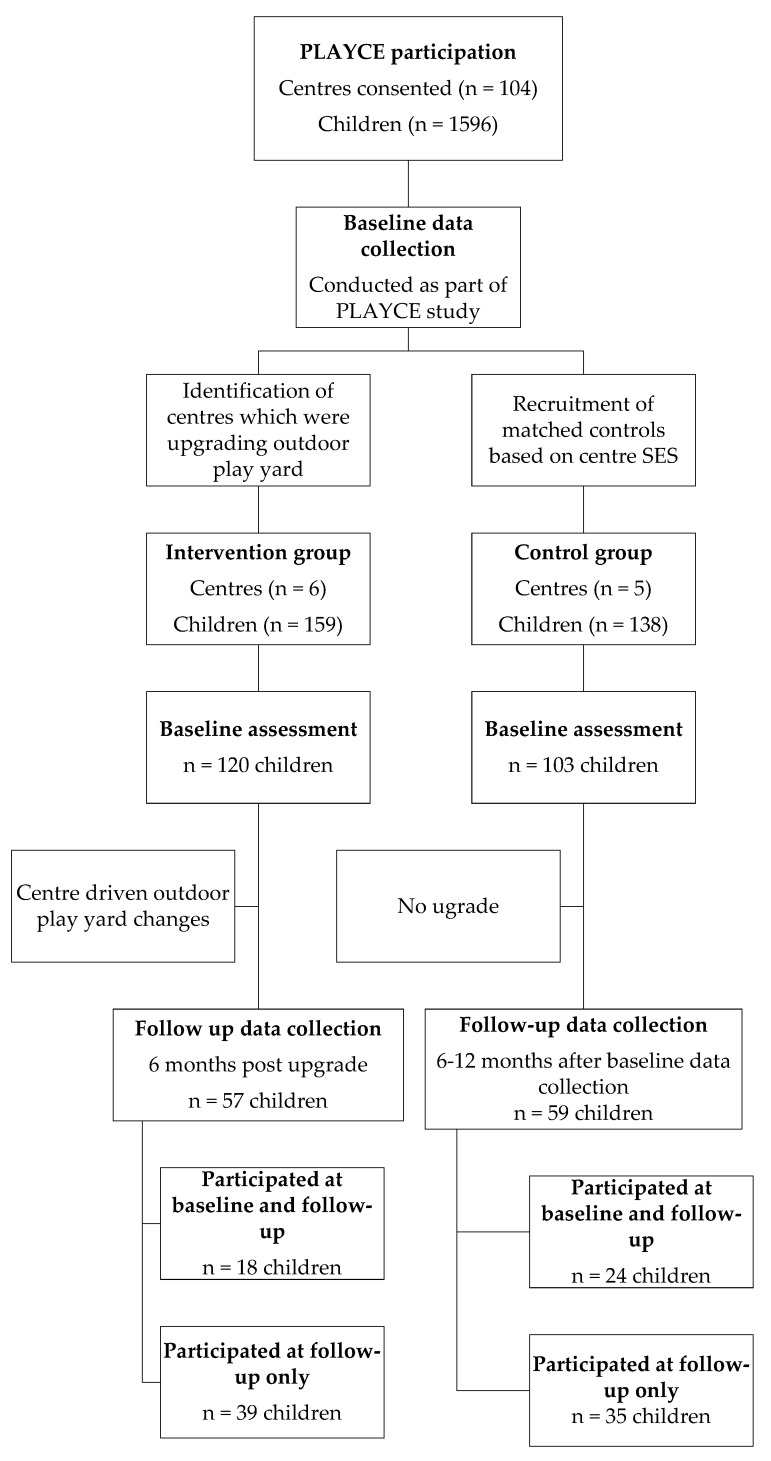
Overview of study design and recruitment process.

**Table 1 ijerph-17-00468-t001:** Child activity levels at baseline and follow-up (total physical activity (TPA) and moderate-vigorous physical activity (MVPA)).

	Intervention (n = 159)			Control (n = 138)			Between Groups Difference (∆ Score) Follow-Up—Baseline		
Variable	Baseline Mean (SD)	Follow-up Mean (SD)	Cohen’s d	Baseline Mean (SD)	Follow-up Mean (SD)	Cohen’s d	Intervention (n = 45)	Control (n = 48)	Cohen’s d
Total physical activity (mins/ECEC day)	58.76 (63.16)	116.85 (40.02)	1.70 ***	68.67 (66.23)	110.81 (41.88)	1.02 ***	84.01 (69.53)	79.81 (56.35)	0.07
Moderate-vigorous physical activity (mins/ECEC day)	28.43 (34.03)	58.88 (24.86)	1.64 ***	34.32 (35.59)	53.49 (26.59)	0.86 ***	44.82 (34.22)	38.99 (28.60)	0.19

TPA: total physical activity; MVPA: moderate to vigorous physical activity; *** *p* < 0.001. Ref = control group.

**Table 2 ijerph-17-00468-t002:** Early childhood education and care (ECEC) outdoor physical environment features by intervention or control centers at baseline and follow-up.

	Intervention (n = 159)			Control (n = 138)			Between Groups Difference (∆ Score) Follow-Up—Baseline		
Variable	Baseline Mean (SD)	Follow-up Mean (SD)	Cohen’s d	Baseline Mean (SD)	Follow-up Mean (SD)	Cohen’s d	Intervention	Control	Cohen’s d
Fixed play equipment ^1^	3.96 (2.44)	1.69 (2.37)	−0.95 ***	3.13 (2.0)	1.67 (2.41)	−0.68 ***	−2.27 (4.36)	−1.47 (3.68)	−0.20
Portable play equipment ^1^	4.87 (3.49)	2.04 (2.78)	−0.90 ***	3.77 (2.42)	2.20 (2.94)	−0.60 ***	−2.83 (5.68)	−1.57 (4.60)	−0.25 *
Natural elements ^1^	4.29 (2.62)	2.03 (2.98)	−0.82 ***	3.43 (2.30)	1.69 (2.26)	−0.77 ***	−2.26 (4.99)	−1.74 (3.91)	−0.12
Outdoor play spaces ^1^	4.34 (3.22)	1.92 (2.94)	−0.79 ***	2.72 (1.82)	1.80 (2.70)	−0.43 ***	−2.42 (5.58)	−0.93 (3.88)	−0.32 ***
Total outdoor playing area ^2^	6.75 (2.57)	6.58 (3.05)	−0.06	4.67 (1.69)	4.95 (3.52)	0.14	0.10 (1.97)	0.75 (3.22)	−0.26

Note. ^1^ Outdoor physical environment subscale score (range from 0 to 10), higher score indicates more activity opportunities; ^2^ Total outdoor playing area rated on scale of 1–10, 0 (no playing area) and 10 (very large area); * *p* < 0.05; *** *p* < 0.001. Ref = control group.

**Table 3 ijerph-17-00468-t003:** Multivariate regression analyses of the association between the ECEC outdoor physical environment, children’s characteristics, and children’s activity levels.

Independent Variables	Standardized Regression Coefficient (β)					
	Model 1 (‘Fixed Play Equipment’)		Model 2 (‘Portable Play Equipment’)		Model 3 (‘Natural Elements’)	
	TPA	MVPA	TPA	MVPA	TPA	MVPA
	15.56 ***	13.04 ***	13.59 ***	11.19 ***	16.49 ***	12.80 ***
Child gender (ref = female)	9.63 *	8.97 **	-	8.42 **	8.85 *	7.79 **
Time (ref = baseline)	−5.63	−0.41	−11.62	−8.79	−3.88	−0.77
Intervention (ref = control)	2.04	3.56	−0.96	−17.91	−6.52	−4.42
Fixed tunnels (ref = not present)	−12.90 **	−12.07 ***	-	-	-	-
Fixed sandbox (ref = not present)	19.80 **	17.89 *** Child age (years)	-	-	-	-
Balls (ref = not present)	-	-	13.43 **	7.84 *	-	-
Twirling equipment (ref = not present)	-	-	−15.38 *	−12.92 *	-	-
Portable slides (ref = not present)	-	-	-	8.41 *	-	-
Portable floor play equipment (ref = not present)	-	-	-	8.16 *	-	-
Real grass (ref = not present)	-	-	-	-	14.46 **	10.04 **
Time x intervention	-	-		12.23 *	-	-

Notes. MVPA: moderate to vigorous physical activity; TPA: total physical activity; * *p* < 0.05, ** *p* < 0.01, *** *p* < 0.001. Model 1: fixed play equipment variables. Model 2: portable play equipment variables. Model 3: natural elements variables. Variables excluded because they were nonsignificant: fixed climbing structures; fixed see-saws; fixed slides; fixed play structure; fixed swings; fixed balancing beams; portable climbing structures; portable jumping equipment; portable floor play equipment (TPA only); portable push/pull toys; portable sand play toys; portable riding toys; portable slides (TPA only); fake grass; other plants; potted plants; vegetation; rocks; flower beds; trees; open areas; water play area; sloping grounds; variety in ground surface; playground built on different levels; size of outdoor space; interaction between time and intervention (except for portable play equipment as independent variable).

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
