# Peer review of "The Effect of Upgrades to Childcare Outdoor Spaces on Preschoolers’ Physical Activity: Findings from a Natural Experiment"

_ijerph, 2020, doi:10.3390/ijerph17020468_

Round 1
Reviewer 1 Report
The manuscript "The Effect of Upgrades to Childcare Outdoor Spaces on Preschoolers’ Physical Activity: Findings from a Natural Experiment" is, to my opinion, contributing with new and interesting knowledge and well-written. It fits well into the journal. In general I enjoyed and appreciated reading it. Relevant and recent literature is included.
This article should definitely be published in the journal.
However, I also have some critical remarks and suggestions:
A main concern is the way how the (descriptive) the results are presented: I would suggest a descriptive table where the activity level for BOTH baseline and follow-up as well as differences (within and between groups) are presented. It is very hard to get an overview over this data based on the presentation-mix (text and table 1). To my opinion the table header “Table 1. Child activity level …” (line 224) does not match what is expressed in the table. Are there really baseline numbers? I don’t think so. What does these numbers really tell? Changes from baseline to follow up, I suppose. This should be addressed explicitly and specifically in the table header. I tried to understand (and recalculate) the “Mean differences” (Between group differences) but didn’t succeed (table 1). What precisely are these differences (follow up? increase?)? I.e: Does this mean that the intervention group increased 58.67 min TPA at baseline with 59.09 min at follow up, so that children in the intervention group at follow-up had 116.85 min TPA? This should be presented in an comprehensive table. (This is a main reason for my suggestion in lit. a). The lack of between group differences in 2 of the 4 subscales (where one could expect differences (->not playing area)) should be more explicitly discussed. Line 217: according to the text the order of numbers in the bracket should be changed (28.43 (…) vs 34.32 (…)), I would say Please reconsider the use of bold letters/fonts in table 1-3, I do not get the “rational” behind. Table 2 (and table 1 regarding between group results): to me it seems a bit counter-intuitive to present negative numbers when reporting “positive changes”
An explicit aim of this particular study, or (an) explicit research question, should be provided (evt. on page 2, after line 93); It is simply lacking. That different children has been studied in the follow-up measurement should have been mentioned earlier, not just in the section “Strengths and limitations” (line 373f) but also in section 2.1. (But it might be that I have misunderstand the term “… recruited retrospectively” Line 82 (line 68-85) : May be it is somewhat “too much” to characterize earlier studies with “discrepancies in results” when just one [30] couldn’t confirm the relation, and the others focus on different aspects of the subject/issue. I would suggest that this reflects too little research based knowledge (but not discrepancies) Figure 1 (p.4): the time span (6-12 months) for the follow up data-collection should at least be commented (if not legitimized). Half a year is a long period in the development of young children’s life. Line 159 ff: Could there be provided any theoretical legitimation or explanation WHY these subscales has been chosen? In general: Is there any theoretical contextualization of the study?
Finally: To me, it is always somewhat annoying when the term “obesity” is the first one meeting the reader (in this case in the abstract), and where the term otherwise by no mean has a crucial impact or stand in the text.
I appreciate very much that the authors broaden their perspective by addressing creative and imaginative play as an important factor that may be related to equipment that does not increase physical activity and, by that, acknowledge different potentials regarding “pedagogical intentions and effects” .
Reviewer 2 Report
The aim of this study is to determine the impact of changes in the physical environment of ECEC outdoors on the physical activity levels of preschool children.
The study is well developed methodologically, although in the analyses shown in tables 1 and 2 the R2 effect sizes are missing.
On the other hand, I am somewhat concerned about the assertions (i.e., line 345) made by the authors in relation to the increase and practice of physical activity of the youngest, since students from 3 to 6 years are still heavily influenced by their parents more than by school. Therefore, I suggest that the authors address this issue in the discussion/conclusions.
Reviewer 3 Report
The manuscript describes the research clearly. My only query is how and why were the cut off points for physical activity chosen? Line 139
Reviewer 4 Report
The manuscript addresses an important topic and is well-written but there are some surprising findings (or they are presenting in ways that are confusing) that warrant further discussion. Most importantly, if I am understanding Table 1 correctly, mean TPA time increased by 42 minutes from baseline to follow-up in the Control children. Yet there is no discussion of why this change would be seen in Controls. Similarly, there are changes in the outdoor physical environment (Table 2) for Controls that are not discussed. Were the Control sites truly unchanged or were there unexpected changes in the environment at those sites as well? Additional comments and questions are detailed below.
The Study design states that control centers were matched one-to-one and yet there are 6 intervention sites and only 5 controls. What happened to the other control site? Are the PA times reported limited to the part of the day when children were at the ECEC? The Instruments section states that children were asked to wear the monitor for at least one weekend day and valid days required 8 hours of wear time. This suggests that data was collected from outside of ECEC time and it is not clear whether all data was used for analyses or only ECEC time. Is there any information about how much time each ECEC allows for outdoor play each day? Since only outdoor time was measured, this may play an important moderating role on the impact of changes. How did Control and Intervention EPAO scores compare at baseline? Were Intervention sites typically lacking in outdoor resources to begin with? From Table 2, all subscale scores got worse at follow-up (other than total outdoor playing area in controls) and yet this is never discussed. It is concerning that ‘upgrades’ (per the title) actually made the outdoor environment worse. The Strengths and Limitations section states that “different children participated at baseline and follow-up.” Can you clarify this statement? Table 1 presents sample sizes for Intervention and Controls but are these only accurate for follow-up measurements? What were the corresponding sample sizes for Intervention and Control groups at baseline?

Reviewer 5 Report
Thank you for your submission, thereby providing me with the opportunity to read your research. This is a well-written paper which is making a justified contribution to the field and has attempted to take a rigorous approach to study design and data collection. I have made a number of comments below for your consideration and response.
Could there be a seasonal variation and effect of weather between baseline and follow-up measurements? Also, could there be a potential impact of having different timescales between baseline and follow-up measurements for intervention and control conditions (i.e., 6 month follow-up for intervention and 6-12 month follow-up for control). Please consider this.
How many researchers collected data using the EPAO? Was accuracy and consistency between members of the research team established? If so, how? Please explain this in the manuscript.
It was stated that there were multiple levels of consent obtained before data collection started. Did the children provide their consent/assent to take part? How were the ethics of researching this group of very young children managed?
Given that the two groups (intervention and control) differed at baseline in terms of TPA, MVPA and % meeting the PA guidelines, could this have an impact upon results and inferences? Essentially, the two groups were not matched at baseline in terms of PA.
Also in terms of inferences, is there something to be said regarding the use of accelerometers to measure PA in ECEC settings. Are the researchers confident that the accelerometers accurately captured the activity intensity of children’s movements (e.g., weight in hands such as crawling in the tunnels).
Can the conclusion please be revised to ensure the main findings of the research are central and explicit.
Please remove ‘kids’ from use in the manuscript.
To avoid any confusion, I suggest that ‘free running’ be removed and alternatively expressed as it is unclear whether the authors are referring to running free or parkour.
